# Buckling Analysis of Laminated Plates with Asymmetric Layup by Approximation Method

**DOI:** 10.3390/ma16144948

**Published:** 2023-07-11

**Authors:** Katarzyna Falkowicz, Pawel Wysmulski, Hubert Debski

**Affiliations:** Department of Machine Design and Mechatronics, Faculty of Mechanical Engineering, Lublin University of Technology, Nadbystrzycka 36, 20-618 Lublin, Poland; p.wysmulski@pollub.pl (P.W.); h.debski@pollub.pl (H.D.)

**Keywords:** composites, critical state, finite element method, thin-walled structures, linear and nonlinear analysis, stability of construction, matrix couplings

## Abstract

This study investigated thin-walled plate elements with a central cut-out under axial compression. The plates were manufactured from epoxy/carbon laminate (CFRP) with an asymmetric layup. The study involved analyzing the buckling and post-buckling behavior of the plates using experimental and numerical methods. The experiments provided the post-buckling equilibrium paths (*P-u*), which were then used to determine the critical load using the straight-line intersection method. Along with the experiments, a numerical analysis was conducted using the Finite Element Method (FEM) and using the ABAQUS^®^ software. A linear analysis of an eigenvalue problem was conducted, the results of which led to the determination of the critical loads for the developed numerical model. The second part of the calculations involved conducting a non-linear analysis of a plate with an initial geometric imperfection corresponding to structural buckling. The numerical results were validated by the experimental findings, which showed that the numerical model of the structure was correct.

## 1. Introduction

Carbon fibre-reinforced polymer composites (CFRPCs) are among the most widely studied lightweight materials. They are widely used in automotive, civil engineering [1,2] and aircraft structures [3,4,5,6,7], among others. Due to their material properties, such as a high strength-to-weight ratio [8,9], these materials are very popular as load-bearing components [8,9,10,11]. Furthermore, their mechanical properties can be shaped by designing specified characteristics and ply configurations for these materials.

Owing to their shape and the fact that they are usually thin-walled, plate elements are particularly susceptible to stability loss [12,13,14,15]. Therefore, the determination of the critical loads for the plates and the analysis of their behavior under dynamic loads are very important parts of their strength analysis, which has been undertaken in a number of studies [12,16,17].

It is worth emphasizing here that there are many numerical and experimental studies on the stability of structures made of classical isotropic engineering materials, and their results are widely reported in the literature. In contrast, there are fewer studies investigating the problem of conjugate buckling for plate structures made of composite materials. Furthermore, there are no experimental studies investigating the buckling and post-buckling behavior of asymmetric laminate plates.

In general, the analysis of layered plates is more complicated due to their anisotropy and heterogeneity. However, with the development of computer techniques and numerical programs, such as FEM, studies attempting to describe these aspects have begun to appear. To give an example, one can mention here studies investigating the performance characteristics for structures made of thin plates, both with open and closed cross-sections, with reinforcements [18,19], initial geometric parameters [20] or cut-outs [21,22]. Special focus was placed on determining the effects of shape inaccuracy [9], the boundary conditions [23], the geometric parameters of the structure [24] or the fibre arrangement [25]. Many of these theoretical considerations were validated through experiments [22]. FEM analyses of the behavior of plates were also conducted by Batoz et al. [26] and Cui et al. [27].

The stability of thin-walled plates made of composite materials was also investigated by Kolakowski and Kowal-Michalska [28], Kolakowski and Krolak [29] and Kubiak [30,31]. These studies predominantly dealt with the problem of buckling in thin-walled composite structures under compression. The above problems were solved by Koiter’s asymptotic theory. Different approaches to stability analysis of laminated composite plates using the Ritz method are presented in papers [32,33,34].

The problem of the stability of compressed thin-walled structures, including mixed-mode buckling, was analyzed using the finite element method by Kubiak [30], as well as by Bazant and Cedolin [35]. Examples of using FEM to solve the problems of the linear and nonlinear stability of composite structures can be found in previous studies conducted by the author [36,37], as well as in studies performed by Alfano and Crisfield [38], Kreja [39], Kopecki [40], Mania et al. [41] and Teter and Kolakowski [42]. Numerical FEM simulations of the behavior of composite structures under different loads were verified experimentally by Debski et al. [43] and Banat et al. [44,45], among others.

The literature review shows that there are numerous studies on the problems of deflection and stability. However, there is a lack of experimental studies on thin-walled asymmetric plates with a cut-out and their use as elastic elements. Furthermore, there are no experimental studies investigating the buckling and post-buckling behavior of asymmetric elements, which is the novel aspect of the current work.

Therefore, the determination of the critical load value that causes the buckling of a thin-walled structure is a very important research problem. The knowledge of this value makes it possible to prevent the structure from premature failure due to a loss of stability by its elements, a problem which was discussed, among others, in [36,46,47,48,49]. It is also worth mentioning that this study involved using the FEM method, which is widely used in many fields [50,51,52,53,54,55,56]. However, the numerically determined critical load value may only yield an approximate estimation of the critical force because the numerical calculations assume an ideal structure, without the geometric imperfections that occur in real structures. This means that the analyzed numerical models of thin-walled structures should be further validated experimentally. To that end, it is necessary to use approximation methods [57,58,59] that enable the estimation of the critical load value based on the experimental results. In this study, the straight-lines intersection method [60,61] was used to estimate the critical forces.

This study investigated the buckling and early post-buckling behavior of compressed thin-walled composite plates with a cut-out. The study involved determining the critical load of a real structure using the straight-lines intersection method, based on the results obtained using the ARAMIS system and analysis of the buckling and post-buckling behavior through the finite element method (FEM). The results showed that this approach was effective for solving the problems of linear and nonlinear stability for thin-walled composite structures.

## 2. Object of the Research

The study was conducted on rectangular thin-walled plates fabricated from an epoxy/carbon composite material (M12/35%/UD134/AS7) through autoclaving [62]. The material properties of the CFRP laminate used for samples were determined experimentally in compliance with the relevant ISO standards, as described in [63]. The mechanical properties of the composite material are presented in Table 1. The plates were subjected to axial compression.

The tested plates had a central rectangular cut-out with variable geometric parameters and constant overall dimensions [64]. A schematic representation of the considered model with its dimensions and ply arrangement is given in Figure 1.

Six plate models with different cut-out dimensions (20 × 100 mm, 30 × 100 mm, 40 × 100 mm, 30 × 120 mm) and three fibre arrangement angles (30°, 45°, 60°) were analyzed. The composite structure consisted of 18 plies; each 0.105 mm thick, in an asymmetric arrangement with respect to the midplane. The considered ply layup is presented in Table 2.

Each plate model consisted of a core spread over the entire plate volume and vertical strips arranged on both sides of the core along the longer edges of the plate (Figure 1b). This ply layup was selected to ensure that flexural-torsional buckling would be the lowest buckling mode of the plate, without additional forcing. The ply orientations were selected using mechanical matrix couplings based on the studies conducted by Ch. York [65,66]. This concept was comprehensively described in the authors’ previous studies [68,69].

## 3. Methodology and Scope of the Study

The range of the conducted study included the analysis of the buckling and early post-buckling behavior of a compressed thin-walled composite plate weakened by cut-outs of various geometrical parameters and different fibre orientations. The study was performed using both experimental and numerical methods. The experiments conducted on the fabricated thin-walled laminate plates enabled the observation of the structure’s behavior in the critical state and after the loss of stability. The numerical simulations conducted in parallel with the experiments were aimed at developing adequate, experimentally validated FEM models for simulating the buckling of a thin-walled laminated structure, closely reproducing the real structure’s behavior.

### 3.1. Experimental

The experiments were carried out on a Zwick/Roell ZMART PRO universal testing machine with a measuring range of up to 2500 kN, coupled with the ARAMIS system [70], which enabled the collection of data in a graphical form and analysis of the deformation and displacement [71]. The experiments were performed with a constant cross-bar velocity of 2 mm/min at room temperature. The axially compressed thin-walled composite plates were loaded with approximately 150% of the numerical critical load value. During the compression process, the plate element was simply supported by specially designed and manufactured grips mounted in the testing machine. The test stand with the mounted plate sample is shown in Figure 2.

During the tests, the compressive force and the shortening of the sample in the perpendicular direction to its cross-section were measured. The shortening of the sample was measured at the top edge of the plate. The ARAMIS optical system was used to measure the plate shortening. This system uses a series of digital images to read the displacements taken during measurements at regular intervals by two cameras positioned at an appropriate distance from the tested object. The cameras are placed on a special tripod. This system is characterized by high resolution and high measurement accuracy. After preparing the sample and setting up the tripod with the cameras, the system was calibrated using a template with marked reference points. After the calibration, the first photo of the sample was taken, which was the zero load state, as well the reference state, against which all calculations were made for the subsequent images. The images were captured in the Start/Mid/Stop Trigger mode until the measurement was completed. After the measurement was completed, in order to start the analysis of the captured images, the surface area (calculation mask) on which the calculations would be carried out was determined based on the size of the facets (Figure 3). Each facet was assigned a unique structure and coordinates, thanks to which they could be recognized in the images captured during the loading process. The last step before the analysis was to manually select a starting point from which the calculation process would begin.

Figure 4 presents a sample frame from a film generated based on the data obtained with the ARAMIS system. By analyzing the frame, it is possible to determine:-the values of the strains/main displacements in the specimen;-changes in the specimen shape during the loading process.

The post-critical equilibrium paths obtained from the measurements, illustrating the relationship between the load and plate shortening, *P-u*, made it possible to determine the critical load value, and thus to evaluate the structure’s performance in the early post-critical range.

### 3.2. Experimental Determination of the Buckling Load

Inaccuracies occurring during the experiments due to different factors—such as boundary conditions and geometric imperfections of the structure, design of the test stand or application of the load—make it difficult to precisely determine the value of the buckling load. Therefore, it is necessary to use approximation methods that enable the estimation of the buckling load value based on the experimental measurements. In this study, the straight-lines intersection method was used to estimate the critical forces [61,72].

The application of the straight-lines intersection method consisted of the approximation of the post-buckling equilibrium path, which describes the relationship between the load and sample shortening measured perpendicularly to the cross-section. To estimate the approximate value of the critical force by solving two linear functions, the post-buckling equilibrium path *P-u* in the early post-critical range—which were determined experimentally—were used. The post-buckling equilibrium paths *P-u* were approximated in selected intervals by two linear functions having the form [72]:(1)$$\left\{\begin{array}{c} P_{cr}=P\frac{a_{1}}{a_{0}}u+P\\ P_{cr}=P\frac{a_{2}}{a_{0}}u+P \end{array}\to \;\left(P_{cr};\;u\right)\right.$$
where *a*_1_, *a*_2_ are unknown function parameters, *P* is the applied load value, *P_cr_* is an unknown critical load value, *u* is the shortening of the plate corresponding to the critical load.

The critical (buckling) load is determined based on the intersection point of the approximation function L1 with the second linear function L2, projected by the horizontal L3 on the coordinate system vertical axis of the post-buckling characteristic of the structure *P-u* (Figure 5). The results obtained using approximation methods are not always unambiguous. The degree of the approximated curve linearity is strictly dependent on the range of data involved in the process of determining the critical loads. In addition, the result significantly depends on the number of points with specific coordinates subjected to the approximation stage.

In this study, the key determinant of the accuracy of the approximation process was the correlation coefficient *R*^2^. This coefficient is used to determine the convergence level between the approximating function and the selected range of the approximated experimental curve. A higher value of the correlation coefficient ensured a higher accuracy of the approximation process. In this experimental approximation of the post-buckling paths, the minimum value of the correlation coefficient was assumed to be *R*^2^ ≥ 0.85.

### 3.3. Numerical Model

The experimental investigation of the stability and post-buckling behavior was conducted in parallel with the numerical modelling using the finite element method. The numerical analysis was performed using the commercial ABAQUS^®^ software. The scope of the analysis included the investigation of the buckling and early post-buckling behavior up to a value of ~150% of the lowest critical load value. The calculations for the critical state included the solution of a linear eigenvalue problem, which led to the determination of the lowest critical load value and the corresponding buckling mode. The maximum potential energy condition was used to solve the eigenproblem. It was solved using the following equation [30]:(2)$$\left(\left[K\right]+\lambda_{i}\left[H\right]\right){\left\{\psi \right\}}_{i}=0$$
where [*K*] is the structural stiffness matrix, [*H*] is the stress stiffness matrix, *λ_i_* is the *i*-th eigenvalue and *ψ* is the *i*-th eigenvector of displacement.

When the {*ψ*}*_i_* value equals zero, this means that the solution is trivial and the structure remains in the initial state of equilibrium. Equation (3) represents the eigenvalue problem, which can help find *n* multiplier *λ* buckling load values and the corresponding buckling mode.
(3)$$|\left[K\right]+\lambda_{i}\left[H\right]|=0$$

In the second part of the study, a nonlinear static analysis was performed. The initial geometric imperfection was flexural-torsional buckling, and it was implemented with an amplitude of 0.1 of the plate thickness. Although nonlinear analyses are carried out with the progressive failure algorithm [67,73,74], this study only focused on the early post-buckling range. In effect, the relationship between the load and column shortening *P-u* in the early post-critical range could be determined. To solve the geometrically nonlinear problem, the incremental-iterative Newton-Raphson method was used.

Numerical model discretization was carried out by means of shell elements with six degrees of freedom at each node. In addition, 8-node shell elements (S8R) with a quadratic shape function and reduced integration were used. More details about the numerical analysis and the discretization process are given in [67,73]. A general view of the numerical model is presented in Figure 6.

The material properties of each layer of the CFRP were the same as those determined experimentally (Table 1). The FEM model characteristics, such as the geometry, the method of load application and the boundary conditions, were adopted as close as possible to those in the experiments (Figure 2).

The boundary conditions for the numerical model reflected a simple support of the compressed composite plate (Figure 4a). The boundary conditions were enforced by constraining the movement of the kinematic degrees of freedom of the nodes located on the top and bottom edges of the plate. The nodes located on the bottom edge had constrained movement of two translational degrees of freedom Uy = Uz = 0 and of one rotational degree of freedom URz = 0, but were allowed free rotation relative to the edge of the plate. The top edge was assigned the same boundary conditions, additionally allowing node displacement in the direction of loading, i.e., Uz = 0 and URz = 0. The vertical edges of the plate remained free during the loading process. Axial compression was applied through uniform loading of the top edge of the plate.

## 4. Results and Discussion

The experiments conducted on the axially compressed thin-walled plates provided information that made it possible to establish a relation between the buckling of the real structures and the external load. The experimental results allowed qualitative and quantitative analyses of the pre-buckling and buckling behavior based on the obtained test parameters. The buckling state was identified based on the obtained buckling mode and its corresponding critical load. The experimental critical loads were used to validate the numerical results.

Examples of the experimental and numerical flexural-torsional buckling modes obtained for the tested plates are shown in Figure 7.

The results show good agreement in qualitative terms and confirm the stable behavior of the tested plates in the post-buckling range. They also confirm that the selected asymmetric configuration with couplings was correct. Examples of the post-buckling flexural-torsional modes obtained for the compressed composite plate with a 40 × 100 mm cut-out and 45° fibre arrangement angle and deflection maps are given in Figure 8.

The measurements of the plate shortening made it possible to determine the post-buckling equilibrium paths, which describe the relationship between the load and shortening, *P-u* (Figure 9). As mentioned above, all samples were tested until failure, according to the progressive failure method, which is described in detail in [67,73,74], where some of the results are reported. However, in the current work, the focus is on the analysis of the buckling state and the determination of the experimental critical forces using the approximation method. Therefore, the behavior in the low post-critical range is sufficient for the analysis (Figure 10). As shown in Figure 10, the experimental results and FEM curves show good agreement in terms of both quantity and quality.

The experimental post-critical equilibrium paths served as a basis for determining the critical load using the straight-lines intersection method. The key problem with this approach is that the measuring range must be selected correctly in order to describe the post-buckling equilibrium path, as this has a direct impact on the results. If the approximation procedures are inappropriate, the experimental critical loads will significantly differ from the numerical values. In addition, for a sufficient compliance of the approximation function with the experimental curve, the approximation range should be selected in such a way as to maintain a high value of the correlation coefficient R^2^ (R^2^ ≥ 0.85).

In the straight-lines intersection method, the post-buckling equilibrium path *P-u* was approximated using two linear functions. The first one was used in the initial interval of the experimental path (buckling), while the other was applied after a visible change in the characteristic (post-buckling). The critical load value was determined as a horizontal line, projecting the intersection point of the approximation functions onto the vertical axis of the diagram (load axis). The critical load values obtained using the straight-lines intersection method are given in Figure 11.

The critical loads obtained using the approximation method were compared with the eigenvalues determined through numerical analysis. The experimental and FEM critical loads are listed in Table 3.

Figure 12 shows a bar chart with the FEM and experimental critical loads obtained for the tested samples. The highest agreement between the experimental and numerical results was obtained for the 45°_40 × 100 plate, while the lowest for the 30°_40 × 100 plate.

It must be remembered that the numerical load values obtained through solving the eigenproblem are upper estimates of the critical load. The agreement between the approximation and numerical critical loads causing a buckling of the thin-walled flat plates weakened by a central cut-out ranges between 0.9% and 7.32% (Table 4). The greatest difference between the experimental and numerical results was obtained for the 30°_40 × 100 mm plate. The smallest difference, of 0.9%, was obtained for the 45°_40 × 100 sample. The average error of the critical loads obtained using the FEM/EXP methods is ~4%, which is an acceptable value as the experimental results represent a lower estimate of the buckling state. The numerical and experimental results show high quantitative agreement, which proves the correctness of the applied approximation method.

## 5. Conclusions

In this study, the behavior of axially compressed thin-walled plate elements weakened by a central cut-out was investigated. An attempt was made to determine the critical load value based on the experimental post-buckling paths obtained using the straight-lines intersection method. The experimental results were then compared with the numerical critical load value determined using the finite element method. The comparison showed high agreement between the experimental and numerical critical loads. This agreement confirms that the proposed procedure can be employed to determine the critical load values for real structures. The correctly determined critical load value of a thin-walled flat plate is of vital importance for operational reasons because it helps prevent structural buckling.

The study has shown that the accuracy of the results strongly depends on the applied approximation parameters. This particularly concerns the selection of an appropriate approximation range and a high value of the correlation coefficient *R^2^* in order to ensure agreement between the experimental structural characteristics and approximation function.

The results provide important information about modelling thin-walled structures made of composite materials. At the same time, they confirm that the numerical models are designed correctly and are thus effective for both eigenproblem calculations and nonlinear static analysis of an early post-buckling response. The results confirm that the numerical model was designed correctly and thus made it possible to simulate the buckling and post-buckling behavior of the compressed plates with a central cut-out.

## Figures and Tables

**Figure 1 materials-16-04948-f001:**
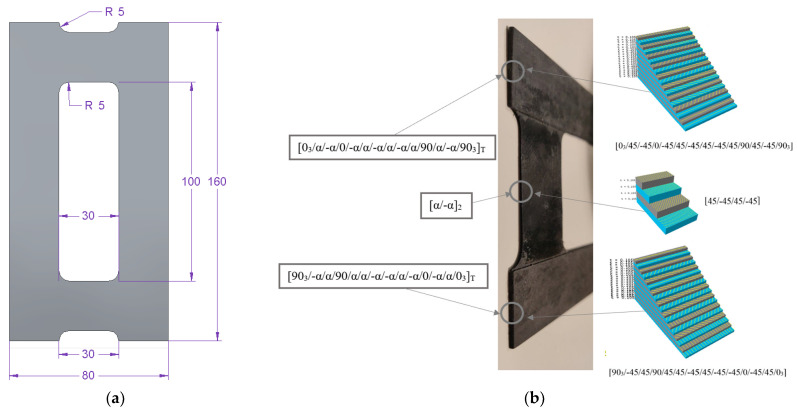
Schematic representation of the considered model: (**a**) geometric model, (**b**) real model with ply arrangement.

**Figure 2 materials-16-04948-f002:**
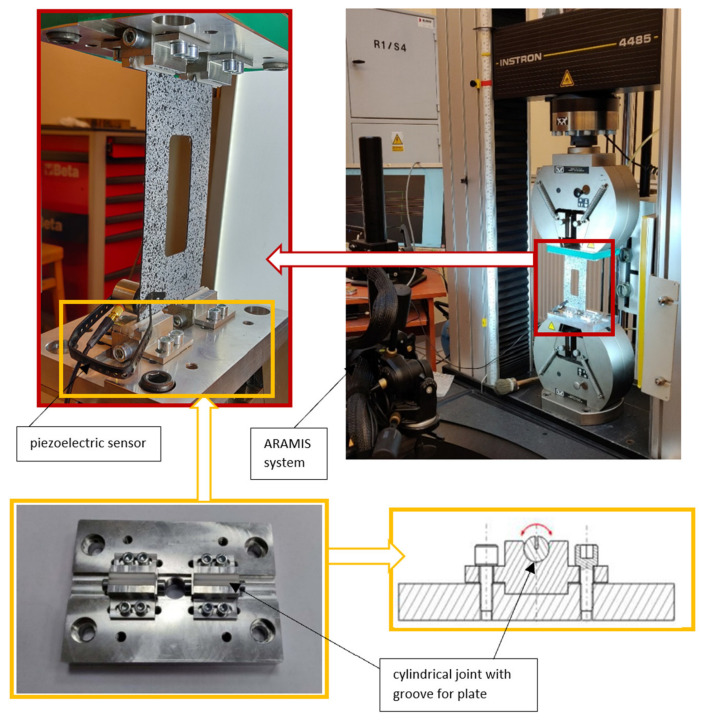
Test stand.

**Figure 3 materials-16-04948-f003:**
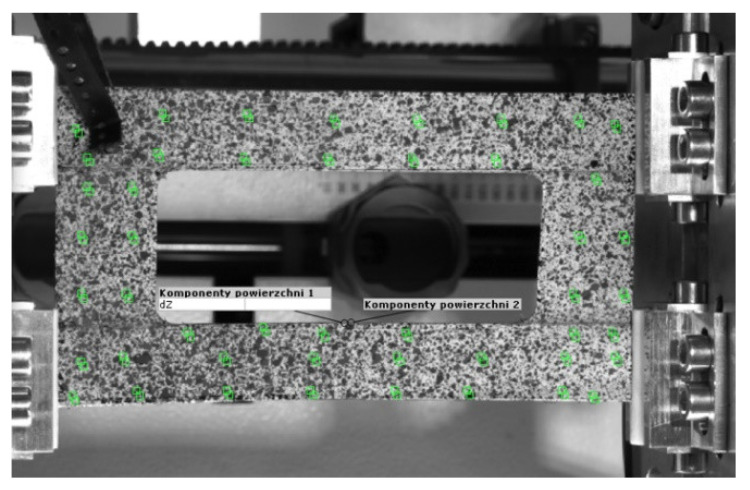
View of the facets in the measuring area.

**Figure 4 materials-16-04948-f004:**
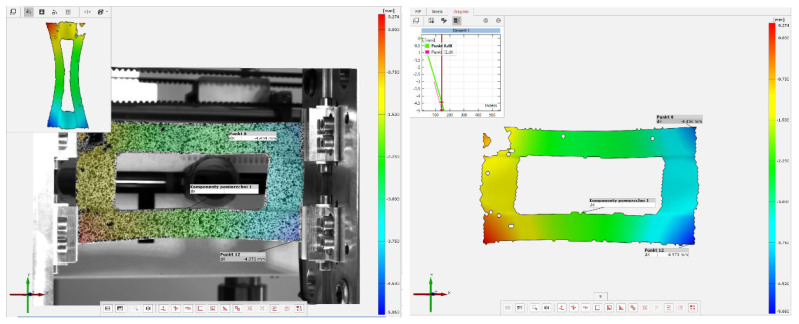
Examples of results generated by the ARAMIS system.

**Figure 5 materials-16-04948-f005:**
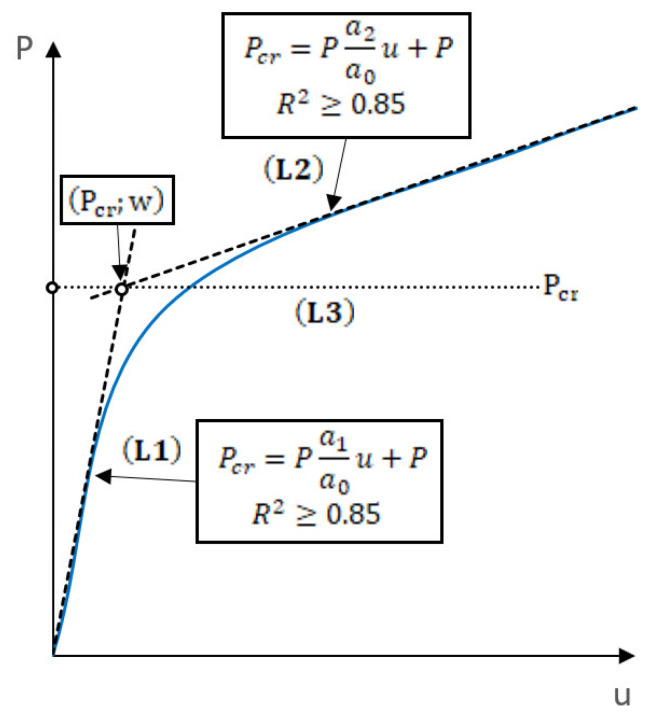
Straight-lines intersection method.

**Figure 6 materials-16-04948-f006:**
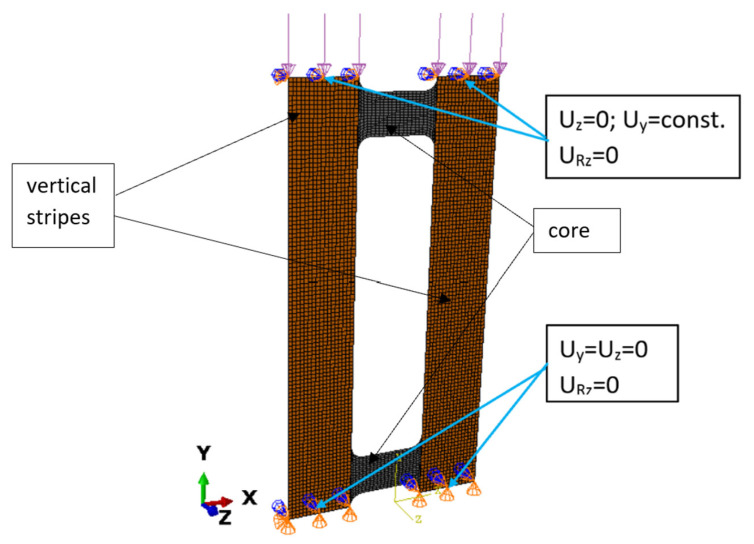
Discrete model of a plate and its boundary conditions.

**Figure 7 materials-16-04948-f007:**
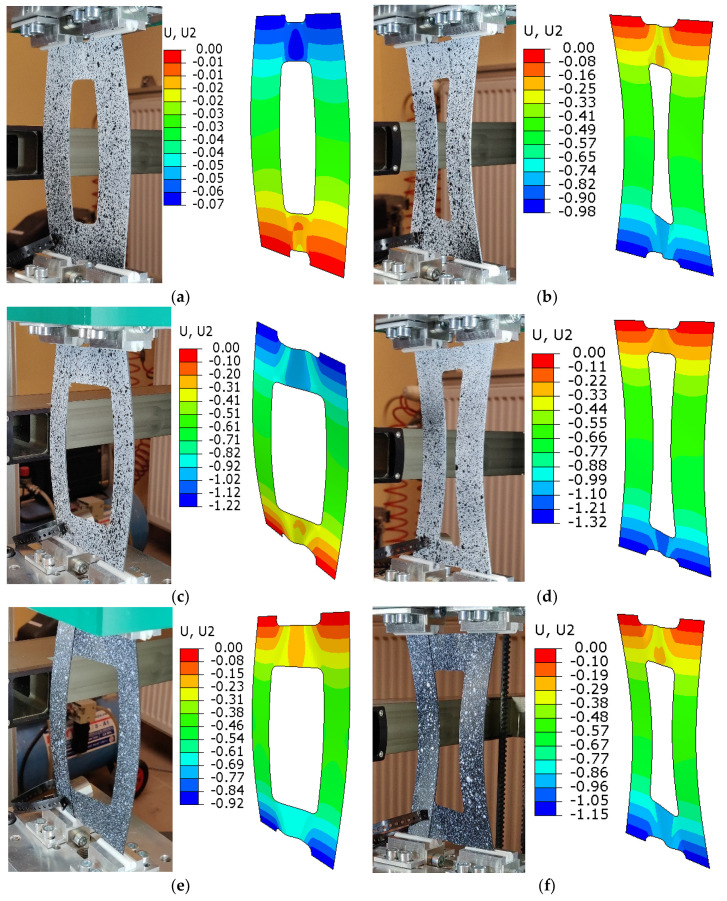
Flexural-torsional buckling modes obtained for the analyzed plates: (**a**) 45°_20 × 100 mm, (**b**) 45°_30 × 100 mm, (**c**) 45°_40 × 100 mm, (**d**) 45°_30 × 120 mm, (**e**) 30°_40 × 100 mm, (**f**) 60°_40 × 100 mm.

**Figure 8 materials-16-04948-f008:**
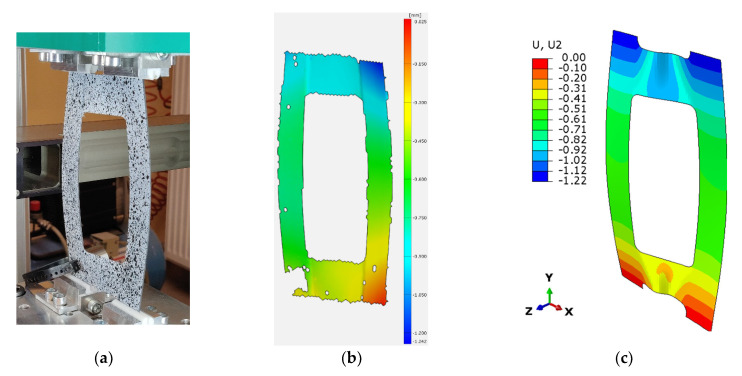
Post-buckling of a plate with a 40 × 100 cut-out and a 45 fibre arrangement angle, together with obtained deflection maps (**a**) experimental, (**b**) ARAMIS, (**c**) Abaqus.

**Figure 9 materials-16-04948-f009:**
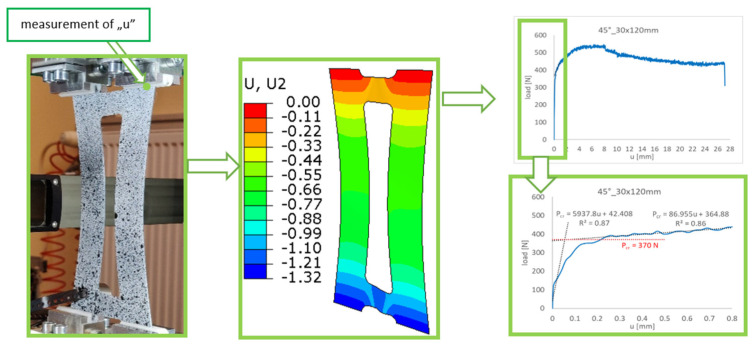
Determination of the critical load from an early post-critical equilibrium path.

**Figure 10 materials-16-04948-f010:**
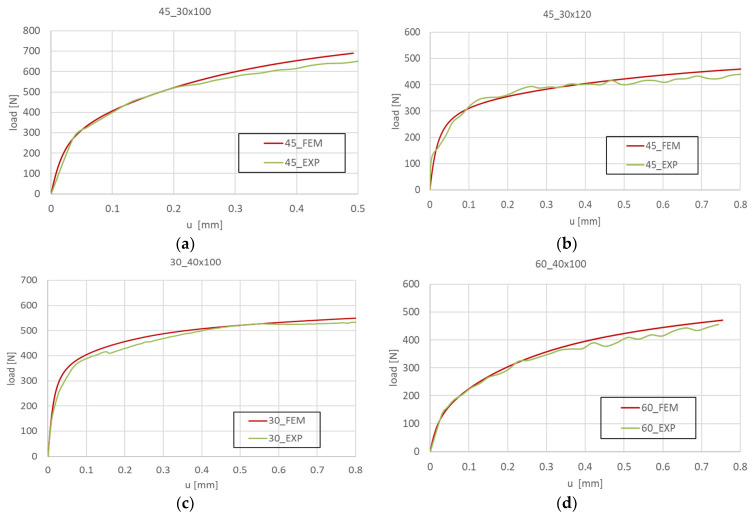
FEM and experimental post-buckling equilibrium paths obtained for the tested plates: (**a**) 45_30 × 100 mm, (**b**) 45_30 × 120 mm, (**c**) 30_40 × 100 mm, (**d**) 60_40 × 100 mm.

**Figure 11 materials-16-04948-f011:**
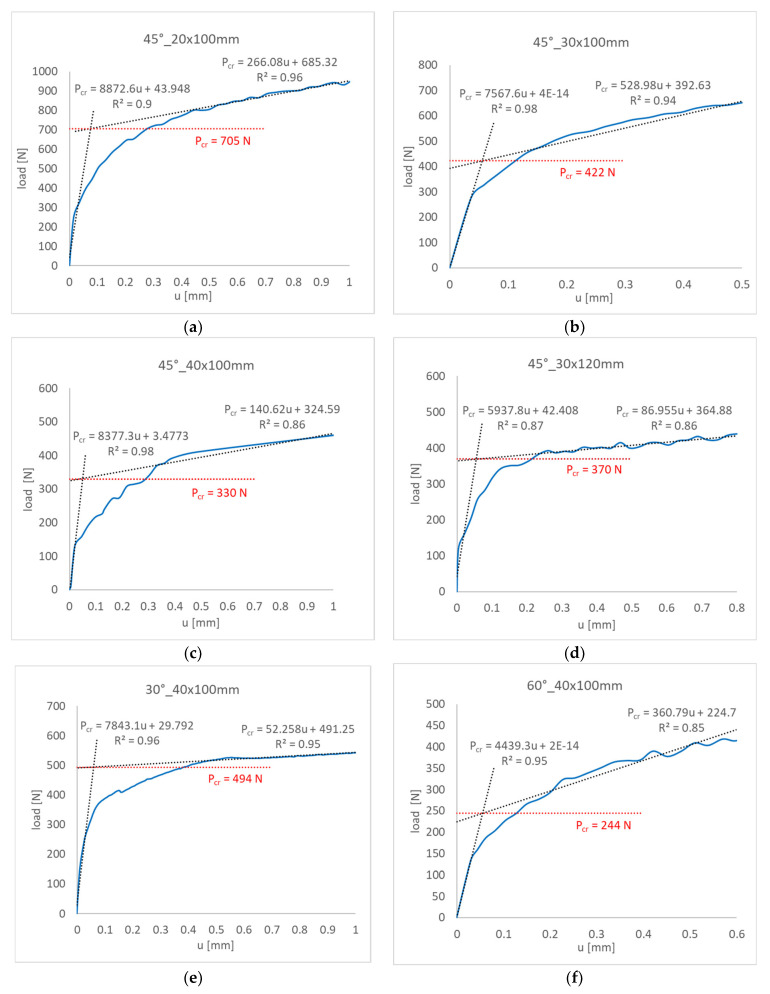
Buckling loads determined for the tested plates by the straight-lines intersection method: (**a**) 45°_20 × 100 mm, (**b**) 45°_30 × 100 mm, (**c**) 45°_40 × 100 mm, (**d**) 45°_30 × 120 mm, (**e**) 30°_40 × 100 mm, (**f**) 60°_40 × 100 mm.

**Figure 12 materials-16-04948-f012:**
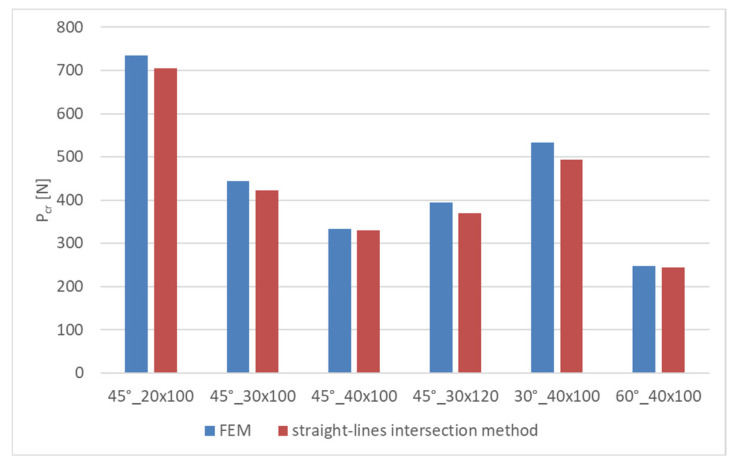
Bar chart showing FEM and experimental critical loads.

**Table 1 materials-16-04948-t001:** Material properties of the tested laminate.

Young’s Modulus [MPa]		Poisson’s Ratio ν_12_	Kirchhoff’s Modulus G_12_ [MPa]
0° (E_1_)	90° (E_2_)	0.36	±45°
143,530	5826		3845

**Table 2 materials-16-04948-t002:** Tested ply configurations.

Ply Number	Core		Strips		Final Configuration	
	Ply Orientation	Coupling	Ply Orientation	Coupling	Configuration	Coupling
18	[α/−α]_2_	A_S_B_t_D_S_ *	[α/−α/0/−α/0/α/90/α/−α]	A_S_B_l_D_S_ *	[0_3_/α/−α/0/−α/α/−α/α/−α/α/90/α/−α/90_3_]T	A_S_B_lt_D_S_

Where: α—angle, A_S_, D_S_—simple laminate, B_t_—extension-twisting and shearing-bending coupling, B_lt_—extension-bending, extension–twisting and shearing–bending coupling, B_t_—extension–twisting and shearing–bending coupling, D_F_—twisting-bending coupling. *—it is some modification of the ply orientation tested by York [65,66], more information about the selection of this configuration was given in [67].

**Table 3 materials-16-04948-t003:** Comparison of FEM and experimental critical loads.

Method	45°_20 × 100	45°_30 × 100	45°_40 × 100	45°_30 × 120	30°_40 × 100	60°_40 × 100
FEM [N]	735	444	333	394	533	247
straight-lines intersection [N]	705	422	330	370	494	244
Difference [N]	20	22	3	24	39	3

**Table 4 materials-16-04948-t004:** Difference [in %] between numerical and experimental critical loads.

Difference [%]	45°_20 × 100	45°_30 × 100	45°_40 × 100	45°_30 × 120	30°_40 × 100	60°_40 × 100
FEM/ straight-lines intersection	4.08%	4.95%	0.9%	6.09%	7.32%	1.21%

## Data Availability

Data are contained within the article.

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
