# Peer review of "Buckling Analysis of Laminated Plates with Asymmetric Layup by Approximation Method"

_materials, 2023, doi:10.3390/ma16144948_

Round 1
Reviewer 1 Report
The work regards the numerical and experimental analysis of the buckling behavior of thin composite plates with cut-outs. Overall, the manuscript is clear and well-written, but some issues require revision:
· Line 101 – Why did you consider this set of dimensions for the samples? Are these dimensions close to those of a specific standard test?
· Lines 103-107 – Explain the technical reasons for choosing three stacking sequences on the different areas of the samples.
· Lines 109-113 – Text formatting is different from the Journal standard.
· Line 138 – Add a reference to paragraph 3.3, where the numerical model is described.
· Line 179 – Briefly describe the theoretical framework behind the straight-lines intersection method. Which is the threshold that determines the onset of elastic instability?
· Line 214 – This statement can be misleading since linear eigenvalue buckling analysis is performed on a theoretically perfect structure, neglecting shape and manufacturing imperfections which decrease the critical buckling load. Hence, the linear buckling analysis provides a theoretical lowest critical load that cannot be reached experimentally.
· Line 226 – Justify the choice of the amplitude factor. Is there any relevant reference to be cited?
· Figure 7 – Are the numerical contours from the linear or non-linear analysis? Also, the displacements measured with the ARAMIS model should be added to the figure to make a quantitative comparison between the FE model and the experiment.
· Considering the wide amplitude of investigations about composite plates, more references should be included in the references as: https://doi.org/10.1016/j.ijmecsci.2020.106094; https://doi.org/10.1016/j.tws.2022.110374; https://doi.org/10.1016/j.ijnonlinmec.2013.06.001
Check the English throughout the manuscript, especially in the Introduction and Abstract paragraphs.
Author Response
I am very appreciated to Reviewer and Editor for their patience and the next chance to improve my manuscript.
I would like to express my sincere thanks for the reviewer's recognition of the work, and I would like to inform you that all suggestions for improving the work quality have been taken into account.

Reviewer 2 Report
Detailed comments for the authors can be found inside the attached PDF file.

Minor improvements to English language are needed.
Author Response
I am very appreciated to Reviewer and Editor for their patience and the next chance to improve my manuscript.
I would like to express my sincere thanks for the reviewer's recognition of the work, and I would like to inform you that all suggestions for improving the work quality have been taken into account.
All suggestions have been corrected in the text.

Reviewer 3 Report
The work is well written, well conducted and well presented. It should definatelly be published as is
Author Response
I am very appreciated to Reviewer and Editor for their patience and for the reviewer's recognition of the work. And I would like to thank the Reviewer for accept manuscript.

Reviewer 4 Report
The manuscript carrying the title "Buckling State Analysis of Laminated Plates with Asymmetric Layer Lay–Up with Approximation Method" presents numerical results besides experimental results for the buckling of laminated plates with a large opening.
The present form of the manuscript cannot be published as its content need to be upgraded.
First the manuscript has to be thoroughly improved regarding the English language.
The innovation of the article should be clearly stated: using the two straight lines method to define the buckling a the plate is an old method already described in the literature. It is known that the plate has a stable post-buckling behavior, therefore the definition of its buckling load is not so important regarding to its structural capability to carry load.
The authors choose to use the end-shortening of the plate as a function of the applied axial load. Why not use the out-of-plane deflection vs. the applied axial load? Please discuss.
Fig. 1 is not clear. Please add a detailed cross section of the plate displaying clearly the layers and their directions.
Please explain the following sentence :"Each plate model consists of a core spread over the entire plate volume and vertical strips arranged on the both sides of the core along to the longer edges of the plate (Fig. 1b)."
Adding a dedicated drawing might give more light to the issue.
Another important issue is the boundary conditions of the specimen in the loading system. Please present what are the theoretical boundary conditions for each side of the tested plate.
Fig. 5 : How many points are involved to approximate the first curve ? The same for the second curve ? Another crucial point is when one stops to add points for the first cure and the same for the second curve. Please explain and discuss the problem.
Table 3 : Are the FEM results based on eigenvalue calculation ?
Were the plates measured for their initial geometric imperfections, prior to the tests ?
Please explain the fluctuations of the plate response in the post buckling regime, as depicted in Fig. 11.
Please provide details about the FE model used in the manuscript: number of elements and nodes and convergence studies performed to make sure the model is adequate for comparison with the experimental one.
The English language is not good. A drastically improvement is required.
Author Response

(The authors gave the same response as above.)

Round 2
Reviewer 1 Report
The required amendments were considered and implemented.
The paper can be accepted in present form.
Reviewer 4 Report
The authors provided adequate answers to my comments and introduced modifications in the manuscript, leading to an acceptable paper.